# Learning to Self-Verify Makes Language Models Better Reasoners

Yuxin Chen [1]  Yu Wang [2]  Yi Zhang [2]  Ziang Ye [2]  Zhengzhou Cai [3]  Yaorui Shi [2]
Qi Gu [4]  Hui Su [4]  Xunliang Cai [4]  Xiang Wang [2]  An Zhang [2]  Tat-Seng Chua [1]

## Abstract

Recent large language models (LLMs) achieve strong performance in generating promising reasoning paths for complex tasks. However, despite powerful generation ability, LLMs remain weak at verifying their own answers, revealing a persistent capability asymmetry between generation and self-verification. In this work, we conduct an in-depth investigation of this asymmetry throughout training evolution and show that, even on the same task, improving generation does not lead to corresponding improvements in self-verification. Interestingly, we find that the reverse direction of this asymmetry behaves differently: learning to self-verify can effectively improve generation performance, achieving accuracy comparable to standard generation training while yielding more efficient and effective reasoning traces. Building on this observation, we further explore integrating self-verification into generation training by formulating a multi-task reinforcement learning framework, where generation and self-verification are optimized as two independent but complementary objectives. Extensive experiments across benchmarks and models demonstrate performance gains over generation-only training in both generation and verification capabilities. Our code is publicly available at https://github.com/chenyuxin1999/Learning-to-Self-Verify.

## 1. Introduction

Large language models (LLMs) have demonstrated strong capabilities in complex reasoning (DeepSeek-AI, 2025;

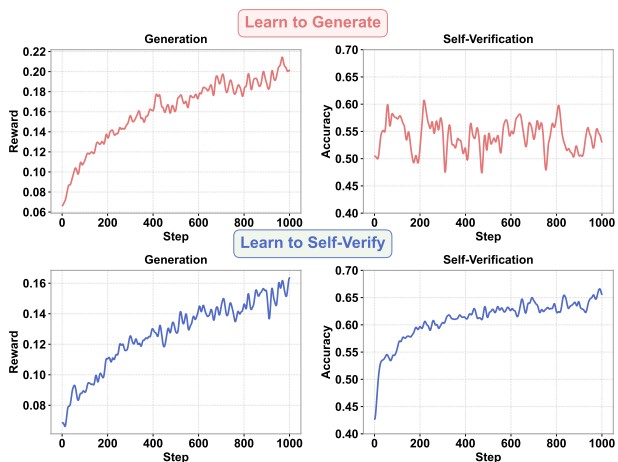

*Figure 1.* Training dynamics of Qwen2.5-1.5B-Instruct. (**Top**) It reveals a persistent asymmetry between generation and self-verification: learning to generate does not lead to improved self-verification ability, even on the same task. (**Down**) In the reverse direction, learning to self-verify not only improves self-verification ability but also leads to improved generation performance.

Yang et al., 2025a; Team, 2025; OpenAI, 2025). With the advancement of Reinforcement Learning with Verifiable Rewards (RLVR), current models have made substantial progress on verifiable tasks such as mathematics and programming (Shao et al., 2025; Anthropic, 2025; Z.AI, 2025), while also showing consistent improvements on open-domain tasks including writing, dialogue, and general problem solving (DeepSeek-AI et al., 2025; MiniMax, 2025; Bhaskar et al., 2025; Zeng et al., 2025b). Despite these advances, a fundamental asymmetry remains: even the most powerful models often lack the ability to reliably verify the correctness of their own outputs.

LLMs have long been considered incapable of verifying the correctness of their own answers (Stechly et al., 2025; Hong et al., 2024; Zhang et al., 2024). With the advent of RLVR, some works observe that models can exhibit emergent self-verification behaviors, sometimes also referred to as an "aha moment" (DeepSeek-AI, 2025; Zeng et al., 2025a; Hu et al., 2025). However, subsequent analyses suggest that most of these behaviors are in fact fake verification: although the model appears to be checking its previous reasoning,

[1]National University of Singapore [2]University of Science and Technology of China [3]Beijing University of Posts and Telecommunications [4]Meituan. Correspondence to: An Zhang <an_zhang@ustc.edu.cn>, Qi Gu <guqi03@meituan.com>.

*Proceedings of the 43rd International Conference on Machine Learning*, Seoul, South Korea. PMLR 306, 2026. Copyright 2026 by the author(s).

this step has little impact on the final answer, fundamentally due to the model's limited ability to reliably verify its own generations (Zhao et al., 2025; Yee et al., 2024). More importantly, self-verification capability does not naturally improve with increased model scale or stronger generation ability (Lu et al., 2025), revealing a persistent asymmetry between generation and self-verification. Motivated by this, several approaches attempt to jointly optimize generation and verification within the same training step, treating verification as an auxiliary component (Liu et al., 2025b; Zhang et al., 2025a; Wang et al., 2025b). In practice, however, the training dynamics of these methods remain dominated by the generation objective, leaving the fundamental asymmetry largely unexplored.

In this work, we conduct an in-depth investigation of the asymmetry between generation and self-verification. Specifically, we explicitly train the LLM to generate better answers in a specific domain (e.g., mathematics) and track how it behaves when verifying its own answers on the same set of tasks throughout training process. We find that this asymmetry still persists: improving a model's generation performance does not lead to corresponding improvements in its ability to verify its own solutions, as illustrated in Figure 1 (top). This naturally raises a key research question: does this asymmetry also manifest in the reverse direction? In other words, *can improving a model's self-verification ability lead to better generation performance*?

To answer this question, we adopt an alternative training paradigm: instead of training the model to generate better answers, we train it solely to judge the correctness of its own solutions. With a carefully designed self-verification training pipeline, we surprisingly find that although training the model for generation does not improve its self-verification ability, training the model to self-verify does improve its generation performance, even achieving comparable performance to standard generation training on several benchmarks, as illustrated in Figure 1 (down). Beyond comparable performance, the resulting models acquire strong verification capability. Benefiting from this improved self-verification ability, we observe a significant reduction in the number of tokens required to solve the same problems, indicating more efficient reasoning. Moreover, stronger self-verification unlocks effective test-time scaling: incorporating self-verification results into majority voting leads to performance gains.

Building on these observations, we further explore integrating self-verification into generation training by formulating a multi-task reinforcement learning framework, where generation and self-verification are optimized as two independent but complementary objectives. Specifically, we introduce two orthogonal training strategies: (i) learning to self-verify as a stronger initial policy before learning to

generate, and (ii) alternating training between generation and verification, where a verification phase is triggered after several generation steps. Extensive experiments show that these integrated training strategies consistently outperform those trained with generation alone.

Our main contributions are as follows:

- We conduct an in-depth investigation of the asymmetry between generation and self-verification throughout training, and show that improving generation ability does not lead to corresponding gains in self-verification.
- We identify the reverse direction of this asymmetry: learning to self-verify can effectively improve generation performance. Based on this insight, we propose to integrate self-verification into generation training by formulating a multi-task reinforcement learning framework.
- We provide extensive experiments demonstrating that learning to self-verify consistently improves problem-solving performance, together with detailed analyses.

We provide a more detailed discussion of our contributions and significance in Appendix B.

## 2. Preliminary

In this section, we introduce the preliminary concepts and notations used throughout the paper. We first review the RLVR formulation in Section 2.1. We then show how the same RLVR framework can be instantiated in two different settings: generation training (Section 2.2), where the model is optimized to solve a given task, and verification training (Section 2.3), where the model is optimized to judge the correctness of a given solution.

### 2.1. RLVR

Reinforcement Learning with Verifiable Rewards (RLVR) is a reinforcement learning framework for training language models using automatically computable reward signals. Instead of relying on human preference models, RLVR employs a rule-based verifier that evaluates each model output against a reference and returns a scalar reward.

Concretely, given an input query $x_i$, where $i$ denotes the query index, the model parameterized by $\pi_\theta$ generates multiple outputs $o_{i,j} = (z_{i,j}, y_{i,j})$, where $j$ indexes different samples generated for the same query. Here, $z_{i,j}$ denotes the intermediate reasoning trace and $y_{i,j}$ denotes the final prediction. A verifier then assigns a reward score $r_{i,j}$ by comparing the model output with a reference solution $y_i^*$. The training objective is to optimize the model parameters so as to maximize the expected verifier reward over model-generated samples:

$$\max_\theta \ \mathbb{E}_{o \sim \pi_\theta(\cdot|x)}[r]. \tag{1}$$

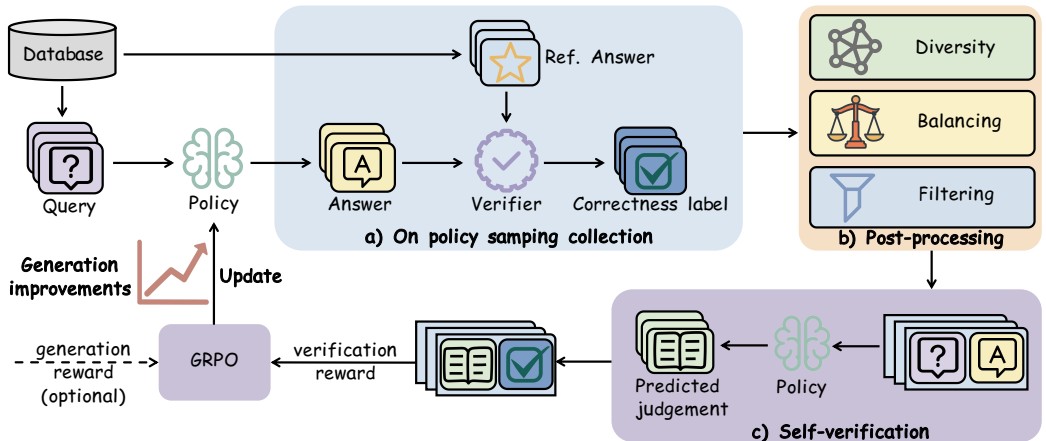

*Figure 2.* Overview of our self-verification training framework. We collect on-policy problem-solving trajectories from the model and obtain correctness labels from a verifier. These trajectories are then processed through a post-processing pipeline, including data balancing, filtering, and diversity-aware sampling, to construct self-verification training data, which is used to train the model to judge the correctness of its own answers. We find that training the model solely for self-verification already leads to improved generation performance. Integrating this self-verification objective into generation training further strengthens the model's generation ability.

In this work, we adopt Group Relative Policy Optimization (GRPO) (Shao et al., 2024) as the underlying optimization algorithm. GRPO can be viewed as a simplified variant of PPO (Schulman et al., 2017) that directly optimizes the policy without introducing a separate value network. For each input $x_i$, the policy samples a set of $G$ candidate outputs $\{(z_{i,j}, y_{i,j})\}_{j=1}^{G}$, each receiving a reward $r_{i,j}$. The policy is then updated by comparing each candidate against the group statistics, using the following clipped surrogate objective:

$$\mathcal{L}_{\text{GRPO}}(\theta) = \mathbb{E}_{x_i \sim \mathcal{D}} \left[ \frac{1}{G} \sum_{j=1}^{G} \frac{1}{|z_{i,j} \circ y_{i,j}|} \right.$$
$$\left. \times \sum_{t=1}^{|z_{i,j} \circ y_{i,j}|} \min \left( \rho_{i,j}^t A_{i,j}, \text{clip}(\rho_{i,j}^t, 1 - \epsilon, 1 + \epsilon) A_{i,j} \right) \right] \quad (2)$$

where:

$$\rho_{i,j}^t = \frac{\pi_\theta(z_{i,j}^t \circ y_{i,j}^t \mid x_i, z_{i,j}^{<t} \circ y_{i,j}^{<t})}{\pi_{\theta_{\text{old}}}(z_{i,j}^t \circ y_{i,j}^t \mid x_i, z_{i,j}^{<t} \circ y_{i,j}^{<t})},$$

where the superscript $t$ denotes the token index in the concatenated sequence, and $\circ$ denotes sequence concatenation. Here, the advantage $A_{i,j}$ is computed by normalizing the rewards within the sampled group:

$$A_{i,j} = \frac{r_{i,j} - \mu_i}{\sigma_i + \epsilon_{\text{norm}}}, \quad (3)$$

with:

$$\mu_i = \frac{1}{G} \sum_{j=1}^{G} r_{i,j}, \qquad \sigma_i = \sqrt{\frac{1}{G} \sum_{j=1}^{G} (r_{i,j} - \mu_i)^2}.$$

Intuitively, GRPO encourages generations that perform better than the group average while suppressing those with

lower relative rewards. Inspired by (Yu et al., 2025a), we adopt the clip-higher strategy and token-level mean advantage normalization.

## 2.2. Generation Training

Under the RLVR formulation, generation training corresponds to the standard task-solving setting. Each training example consists of a task query $x_i$ and a reference solution $y_i^*$. Given $x_i$, the model samples multiple candidate solutions $\{(z_{i,j}, y_{i,j})\}_{j=1}^{G}$, and the verifier assigns a reward by checking whether each generated answer $y_{i,j}$ matches the reference solution $y_i^*$. In this case, the reward signal directly reflects task-solving correctness, and RLVR reduces to optimizing the policy to produce correct solutions for the given tasks.

## 2.3. Verification Training

Under the same RLVR formulation, verification training corresponds to a different instantiation of the input and reference. Each training sample is constructed from a triplet $(x_i, y_{i,j}, c_{i,j})$, where $x_i$ is the task query, $y_{i,j}$ is a candidate solution generated by the model, and $c_{i,j} \in \{0, 1\}$ is a binary correctness label indicating whether $y_{i,j}$ matches the reference solution $y_i^*$. Given such an input $(x_i, y_{i,j})$, the model is prompted to output a judgment $\hat{c}_{i,j}$ indicating whether the provided solution is correct. A rule-based verifier then assigns a reward by comparing the model's judgment $\hat{c}_{i,j}$ with the reference label $c_{i,j}$. In this setting, the model is not optimized to solve the task itself, but rather to assess the correctness of given solutions.

*Table 1.* Evaluation of learning to self-verify across six mathematical reasoning benchmarks. We report both task accuracy (Acc@16 ↑) and average reasoning length in tokens (Tokens ↓) for each model trained under two different objectives: train LLM to generate better solutions (Generate), and train LLM to verify its own solutions (Self-Verify). Results show that models trained with self-verification yield efficient reasoning traces, while achieving comparable or sometimes even better performance than models trained for generation.

| Method | AMC23 | | Minerva | | Olympiad | | Math500 | | AIME24 | | AIME25 | | Avg | |
|---|---|---|---|---|---|---|---|---|---|---|---|---|---|---|
| | Tokens↓ | Acc↑ | Tokens↓ | Acc↑ | Tokens↓ | Acc↑ | Tokens↓ | Acc↑ | Tokens↓ | Acc↑ | Tokens↓ | Acc↑ | Tokens↓ | Acc↑ |
| **Qwen2.5-1.5B-Instruct** | | | | | | | | | | | | | | |
| Generate | 1402 | 30.5 | 963 | 12.4 | 1639 | 20.6 | 936 | 53.7 | 2580 | 2.9 | 2103 | 0.8 | 1604 | 20.2 |
| Self-Verify | 1309 | **33.0** | 870 | **14.0** | 1351 | **22.2** | 817 | **54.6** | 1467 | **4.8** | 1545 | **1.3** | 1227 | **21.7** |
| **Qwen2.5-3B-Instruct** | | | | | | | | | | | | | | |
| Generate | 2754 | **50.9** | 2006 | 17.0 | 3299 | 27.4 | 2021 | 59.6 | 4811 | 8.1 | 4744 | **8.1** | 3273 | 28.5 |
| Self-Verify | 1825 | 46.7 | 1658 | **17.1** | 1891 | **32.1** | 1237 | **65.6** | 2755 | **9.2** | 2252 | 6.3 | 1936 | **29.5** |
| **Qwen2.5-7B-Instruct** | | | | | | | | | | | | | | |
| Generate | 3967 | **65.3** | 2353 | **25.3** | 4543 | 37.8 | 2437 | 70.9 | 7053 | 16.3 | 6397 | **18.1** | 4458 | **38.9** |
| Self-Verify | 1194 | 59.7 | 823 | 25.2 | 1168 | **39.8** | 783 | **74.7** | 1575 | **19.4** | 1369 | 11.7 | 1152 | 38.4 |

## 3. Learning to Self-Verify

In this section, we investigate the reverse direction of this long-standing asymmetry: whether a model can improve its generation performance solely by learning to verify its own solutions. We first introduce our self-verification training pipeline in Section 3.1, then describe the experimental setup in Section 3.2, present the main results in Section 3.3, and provide further analysis in Section 3.4. Additional experiments on broader domains, model families, ablations, on-policy data, and repeated runs are provided in Appendix A.

### 3.1. Self-Verification Framework

Following the notation in Section 2, we now introduce our self-verification framework, as illustared in Figure 2.

**On-Policy Sample Collection**    At each training iteration, we sample a mini-batch of $B$ queries $\{x_i\}_{i=1}^{B}$. For each query $x_i$, we use the current policy $\pi_\theta$ to generate $G$ candidate answers, resulting in $B \times G$ generated samples. For each generated sample, the model produces a solution $y_{i,j}$ together with its corresponding reasoning trace $z_{i,j}$, where $j = 1, \ldots, G$. A rule-based verifier then compares each $y_{i,j}$ with the reference answer $y_i^*$ and assigns a binary correctness label $c_{i,j}$. Each sample is thus represented as a triplet $(x_i, y_{i,j}, c_{i,j})$. All such triplets are stored in a temporary buffer and serve as the raw candidates for constructing the self-verification training data.

**Post-Processing**    At each iteration, the on-policy sampling procedure produces $B \times G$ samples. Directly using all of them for verification training is computationally expensive and can also introduce instability due to imbalance or low-quality samples. For a fair comparison, we downsample these candidates and construct a verification training batch

of size $B$ by selecting the most informative samples. Specifically, we apply the following steps:

- **Filtering:** We first discard invalid samples, including those with malformed outputs, excessively long generations, or missing a unique final answer. We further discard queries for which all generated answers are incorrect, as such cases typically exceed the current capability of the model and provide little useful supervision signal for self-verification.

- **Diversity Control:** To avoid overfitting to a small subset of queries when conducting self-verification training, we perform sampling at the query level and ensure that the selected verification samples are drawn from diverse input queries.

- **Data Balancing:** Since generation often produces highly imbalanced labels (e.g., mostly incorrect at early stages and mostly correct at later stages), while self-verification is essentially a binary classification task, we explicitly enforce each mini-batch of verification data to contain an equal number of correct and incorrect samples.

We ablate these post-processing components in Appendix A.2, showing that filtering and label balancing are important for making the verification objective well-posed. Appendix A.3 further compares on-policy and off-policy verification data, showing that verifying the model's own rollouts is important for effective self-verification training.

**Training**    In self-verification training, the model is prompted with a query–answer pair $(x_i, y_{i,j})$ and is required to predict whether the provided answer is correct. Let $\hat{c}_{i,j}$ denote the model's predicted judgment and $c_{i,j}$ the reference correctness label obtained from the rule-based verifier.

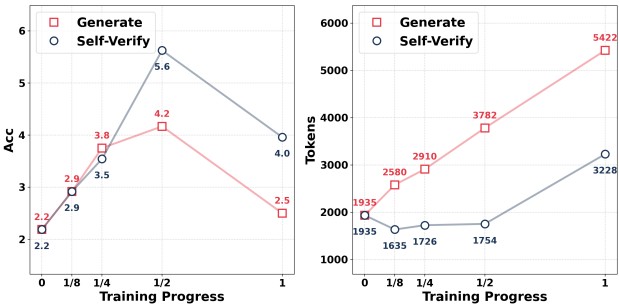

*Figure 3.* Comparison of accuracy and token usage between generation training and self-verification training on AIME24 with Qwen2.5-1.5B-Instruct.

A verification reward is then computed as:

$$r_{i,j}^v = \text{Verifier}(\hat{c}_{i,j}, c_{i,j}). \tag{4}$$

We then optimize the model using the same GRPO objective as in Section 2.1. This training stage treats the model purely as a verifier and encourages it to improve its ability to distinguish correct from incorrect answers. We emphasize that at this stage, the training objective contains no generation reward. The policy is optimized solely to maximize the expected verification reward.

### 3.2. Experimental Setup

We conduct extensive experiments to compare the effects of training LLMs to generate solutions and training them to self-verify. We first describe our experimental setup.

**Dataset and Benchmarks**  For training, we use DAPO-Math-17K (Yu et al., 2025a), a dataset widely adopted for mathematical reasoning. We evaluate our models on six challenging mathematical reasoning benchmarks: AIME24 (Zhang & Math-AI, 2024), AIME25 (Zhang & Math-AI, 2025), AMC23, Minerva, MATH500 (Lightman et al., 2024), and OlympiadBench (He et al., 2024).

**Implementation**  We choose Qwen2.5-1.5B-Instruct, Qwen2.5-3B-Instruct, and Qwen2.5-7B-Instruct as backbone models (Yang et al., 2024). We use veRL (Sheng et al., 2025) as the training framework to implement our RL-based methods with a rule-based verifier. For both generation training and self-verification training, we train the models for 1000 steps. For fair comparison, both generation and verification training use a batch size of 128 with a group size of 8. We set the maximum generation length to 10,240 tokens for all models, with the temperature set to 0.6 and top-$p$ to 0.95.

**Evaluation**  We compare models trained exclusively for generation with those trained exclusively for self-verification on the benchmarks. We report two main metrics:

*Table 2.* Evaluation of verification capability. We report the Acc@8 of different models in judging the correctness of solutions generated by DeepSeek-R1-Distill-Qwen-7B.

| Model | Base | Generate | Self-Verify |
|---|---|---|---|
| Qwen2.5-1.5B-Instruct | 45.58 | $45.95^{+0.37}$ | $\mathbf{62.31}^{+16.73}$ |
| Qwen2.5-3B-Instruct | 59.82 | $55.19^{-4.63}$ | $\mathbf{65.69}^{+5.87}$ |
| Qwen2.5-7B-Instruct | 64.46 | $68.84^{+4.38}$ | $\mathbf{69.50}^{+5.04}$ |

(1) **Acc**, measured by Avg@16 accuracy, (2) **Token**, calculated as the average number of tokens (including both intermediate reasoning and the final answer) across all outputs on each test set. This metric reflects the reasoning efficiency of the model.

### 3.3. Main Results

Table 1 summarizes the performance and reasoning length across six benchmarks and three models, comparing models trained solely to generate answers with models trained solely to judge the correctness of their own solutions. In addition, Figure 3 illustrates the evolution of accuracy and token usage throughout training on AIME24 with Qwen2.5-1.5B-Instruct. From the results, we can draw two key conclusions:

**Learning to self-verify achieves comparable performance to learning to generate.**  Across all models and datasets, training the model solely for self-verification yields performance that is comparable to, and in some cases better than, that achieved by generation-only training. For example, for Qwen2.5-1.5B-Instruct, the self-verification-trained model outperforms the generation-trained model in accuracy across all benchmarks. For Qwen2.5-3B-Instruct, self-verification achieves 32.1% accuracy on OlympiadBench and 65.6% on Math500, surpassing the generation baseline by 4.7% and 6.0%, respectively, demonstrating strong potential even without explicit generation training. This points to an interesting asymmetry in the reverse direction: while improving a model's generation performance does not lead to a corresponding improvement in its ability to self-verify, even on the same task (*cf.* Figure 1), improving self-verification alone can in turn enhance generation performance.

**Learning to self-verify requires significantly fewer tokens to solve the same problems.**  Across all models and datasets, training the model solely for self-verification consistently produces much shorter reasoning traces than generation-only training, while maintaining comparable performance. Notably, for Qwen2.5-7B-Instruct, the self-verification-trained model achieves performance comparable to generation training using only about 25% of the tokens. For Qwen2.5-3B-Instruct, it uses roughly 60% of the tokens while even slightly outperforming the generation baseline. These results indicate that, although the final

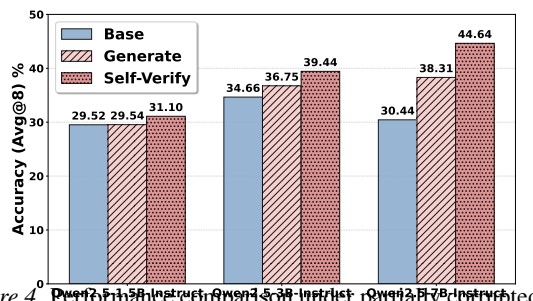

*Figure 4.* Performance comparison under partially corrupted reasoning prefix setting.

*Table 3.* Test-time scaling with self-verification with Qwen2.5-1.5B-Instruct. We report Acc@32 and compare standard majority voting and majority voting augmented with self-verification across three training regimes: Base, Generate, and Self-Verify.

| Method | AIME25 | MATH500 | Olympaid | Minerva |
|---|---|---|---|---|
| **Base** | | | | |
| Major voting | 3.30 | 52.20 | 22.40 | 14.30 |
| + Self-verify | $3.30^{+0.00}$ | $52.40^{+0.20}$ | $22.30^{-0.10}$ | $10.70^{-3.60}$ |
| **Generate** | | | | |
| Major voting | 0.00 | 54.20 | 23.40 | 15.80 |
| + Self-verify | $0.00^{+0.00}$ | $53.00^{-1.20}$ | $23.60^{+0.20}$ | $13.60^{-2.20}$ |
| **Self-Verify** | | | | |
| Major voting | 3.30 | 55.20 | 25.80 | 16.20 |
| + Self-verify | $\mathbf{6.70}^{+3.40}$ | $\mathbf{56.40}^{+1.20}$ | $\mathbf{27.20}^{+1.40}$ | $\mathbf{16.20}^{+0.00}$ |

performance is comparable, self-verification leads to substantially more efficient reasoning traces. We attribute this to the strengthened self-verification ability induced by our training, which enables the model to better recognize when its current solution is likely incorrect and when verification should be triggered. As a result, the model avoids redundant or "fake" verification behaviors and follows more direct solution trajectories. This markedly different reasoning behavior further motivates us to regard self-verification and generation as complementary training signals.

### 3.4. Analysis

In this section, we conduct a detailed analysis to investigate what capabilities are acquired by learning to self-verify and how these capabilities can be exploited in practice. Based on our experiments, we make the following observations:

**Explicit self-verification training turns the model into a strong verifier.** Figure 1 demonstrates that even models with limited parameter sizes can verify their own solutions much more accurately after self-verification training. To further evaluate the model's verification capability as a general verifier in specific domains, we construct a verification evaluation set consisting of benchmark solutions generated by DeepSeek-R1-Distill-Qwen-7B. The model is then asked to judge whether each solution is correct or incorrect, and the results are reported in Table 2. The results show that, since self-verification is essentially a verification task, our model naturally acquires the ability to assess solutions produced by other models as well, demonstrating strong general-purpose verification capability. In contrast, models trained only for generation overall achieve marginal performance gain and in some cases even exhibit noticeable degradation.

**Learning to self-verify enables the model to identify and correct errors in its reasoning process.** We observe that after self-verification training, the number of tokens required to solve a problem is significantly reduced. This suggests that the model starts to precisely trigger verification when it detects potential errors in its reasoning and to correct them in time, thereby avoiding redundant or "fake" ver-

ification behaviors. To validate this hypothesis, we construct a dedicated evaluation set to assess the effectiveness of self-verification behaviors during the reasoning process. Specifically, we mix data from different benchmarks and build a set of 1,545 problems. We first collect the original reasoning trajectories generated by Qwen2.5-7B-Instruct on these problems. We then use GPT-4.1 (OpenAI, 2023) to randomly rewrite these trajectories into reasoning step prefixes with varying numbers of steps and injected some mistakes. The model is prompted with the original query and the corrupted prefix, and is asked to continue the reasoning process. Under this setting, a higher success rate indicates that the model is more capable of detecting errors in the ongoing reasoning and correcting them through effective self-verification. As shown in Figure 4, the self-verification-trained model significantly outperforms both the base model and the generation-trained model, demonstrating substantially stronger error detection and correction capability during reasoning. In contrast, generation training yields only marginal improvements over the base model in this setting.

**Effective self-verification enables test-time scaling.** With a substantially improved self-verification capability, the model can reliably assess the correctness of its own candidate solutions, which unlocks a new form of test-time scaling based on self-verification. Specifically, at inference time, we sample multiple candidate solutions, let the model verify each of them, and aggregate the verification results to obtain a verification score for each candidate. We then jointly consider the majority vote and the verification scores to determine the final answer. Experimental results in Table 3 show that introducing this additional self-verification signal at test time consistently improves performance, demonstrating that self-verification provides an effective and principled way to scale inference beyond naive sampling or self-consistency.

We further validate this trend on general reasoning, code

*Table 4.* Evaluation of integrating self-verification into generation training across six mathematical reasoning benchmarks. We report task accuracy (Acc@16 ↑) for each model under four training strategies. Results show that our strategies improve overall performance over standard and mixed training.

| Method | AMC23 | Minerva | Olympiad | Math500 | AIME24 | AIME25 | Avg |
|---|---|---|---|---|---|---|---|
| **Qwen2.5-1.5B-Instruct** | | | | | | | |
| Generate | 30.5 | 12.4 | 20.6 | 53.7 | 2.9 | 0.8 | 20.2 |
| Mixed-Train | 33.3 | 12.7 | 21.2 | 53.8 | 3.8 | 1.5 | 21.1 |
| Verify-Init | 33.0 | 13.0 | 21.9 | **54.7** | **5.4** | 1.3 | 21.6 |
| Verify-Alter | **36.4** | **13.9** | **22.4** | 54.2 | 5.0 | **4.2** | **22.7** |
| **Qwen2.5-3B-Instruct** | | | | | | | |
| Generate | **50.9** | 17.0 | 27.4 | 59.6 | 8.1 | **8.1** | 28.5 |
| Mixed-Train | 49.4 | 17.6 | 27.5 | 59.2 | **10.8** | 6.3 | 28.5 |
| Verify-Init | 47.8 | 18.3 | 29.5 | 63.1 | 9.6 | 6.5 | 29.1 |
| Verify-Alter | 47.7 | **18.7** | **30.2** | **64.5** | 9.4 | 5.6 | **29.4** |
| **Qwen2.5-7B-Instruct** | | | | | | | |
| Generate | 65.3 | 25.3 | 37.8 | 70.9 | 16.3 | **18.1** | 38.9 |
| Mixed-Train | 59.2 | 24.6 | **40.5** | 73.5 | 15.8 | 9.8 | 37.2 |
| Verify-Init | 63.6 | **26.0** | 39.0 | **74.0** | 17.3 | 12.5 | 38.7 |
| Verify-Alter | **68.0** | **26.0** | 39.0 | 72.9 | **18.3** | 17.7 | **40.3** |

generation, Llama3.2 models, and Qwen2.5-Math in Appendix A.1.

# 4. Integrating Self-Verification into Training

We observe that training a model solely to verify its own answers already improves its generation performance to a level comparable with models trained purely for generation, while exhibiting a markedly different inference behavior: verification-only models produce significantly shorter outputs, indicating a more efficient reasoning trace. This markedly different reasoning behavior further motivates us to view self-verification and generation as complementary training signals. Building on this observation, we further propose to integrate self-verification into generation training.

## 4.1. Multi-Task RL Pipeline

In this work, we formulate the integration of generation and self-verification as a multi-task reinforcement learning problem, where the two objectives are optimized in a decoupled manner. Under this framework, we consider two simple yet effective strategies that are orthogonal: a stage-wise initialization strategy and an alternating training strategy.

**Stage-wise Initialization** We first train the model with a self-verification objective by optimizing the policy to maximize the verification reward $r_v$, as described in Section 3.1. The resulting model, which already possesses a stronger ability to judge the correctness of its own outputs, is then used as a better initial policy for standard generation training, where the policy is further optimized to maximize the

generation reward $r_g$.

**Alternating Training** We alternate between generation training and self-verification training. Specifically, we run generation training for $n$ steps to optimize the policy with respect to the generation reward $r_g$. Every $n$ generation steps, we trigger a self-verification phase, during which the same policy is optimized with respect to the verification reward $r_v$, using the answers generated in the preceding generation phase to construct verification training data. This process is repeated throughout training, allowing the policy to be continuously shaped by both objectives.

In both strategies, generation and self-verification are optimized under the same RLVR framework using GRPO, and the only difference lies in which reward signal ($r_g$ or $r_v$) is used at each stage of training.

## 4.2. Experimental Setup

**Baseline** To benchmark the effectiveness of our method, we compare it against two primary baselines. *Generate* follows the standard RL-based training paradigm for reasoning models and optimizes the policy solely with respect to the generation reward. *Mixed-Train* (Zhang et al., 2025a) jointly optimizes generation and self-verification objectives within each training step. For fair comparison, all baselines and our methods are trained using the same implementation and the same set of hyperparameters.

**Implementation and Evaluation** We use the same datasets, model architectures, implementation details, and evaluation protocols as in Section 3.3. In this section, we evaluate two strategies for integrating self-verification into

training: *Verify-Init*, which corresponds to the stage-wise initialization strategy, and *Verify-Alter*, which corresponds to the alternating training strategy. For *Verify-Init*, we initialize the model from a checkpoint obtained after 400 steps of self-verification-only training, and then further train it for 600 steps with the generation objective. For *Generate*, *Mixed-Train*, and *Verify-Alter*, we train the models for 1000 steps in total.

### 4.3. Results and Analysis

Table 4 summarizes the performance across six benchmarks and three models. Beyond training the model solely for self-verification, we find that integrating self-verification into generation training consistently leads to improved generation performance across most models and benchmarks. Compared to *Mixed-Train*, which directly mixes the two objectives within a single optimization step, our framework decouples the optimization of generation and verification and optimizes them in a coordinated but separate manner, demonstrating additional performance gains. For instance, for Qwen2.5-1.5B-Instruct, *Verify-Alter* improves the average accuracy from 20.2% to 22.7%, outperforming both standard generation training and mixed-objective training. Notably, on AMC23, it improves accuracy by 5.9 points, and on the more challenging AIME benchmarks, it raises accuracy from 0.8% to 4.2%. This suggests that self-verification provides a complementary and beneficial training signal.

We provide additional discussion on why decoupled optimization can outperform direct mixed-objective training in Appendix A.5. We also report repeated-run results in Appendix A.4, confirming that the improvement of Verify-Alter is stable across independent trials.

## 5. Related Works

### 5.1. LLM as Generator

Improving the generation capability of LLMs has long been a central focus of the community (Brown et al., 2020; OpenAI, 2023). Early works typically collect high-quality trajectories with complex reasoning patterns and train LLMs via imitation learning (Ouyang et al., 2022; Touvron et al., 2023; Yang et al., 2024). With the success of models such as DeepSeek-R1 (DeepSeek-AI, 2025), which is trained with the GRPO algorithm (Shao et al., 2024), a surge of follow-up research has been inspired. Meanwhile, Reinforcement Learning with Verifiable Rewards (RLVR) has emerged as a powerful and scalable training paradigm for further boosting LLM generation performance by leveraging verifiable reward signals (Jin et al., 2025; Wang et al., 2025f). Building on these foundations, more recent studies start to investigate how to extend LLMs' advanced generation ability to broader domains (Su et al., 2025; Yu et al., 2025c; Gunjal

et al., 2025), make generation more efficient at inference time (Sui et al., 2025; Feng et al., 2025; Wang et al., 2025a), and improve stability and effectiveness of generation training (Yang et al., 2025c; Wu et al., 2025; Chen et al., 2025d). These advances have led to a series of increasingly capable large models (OpenAI, 2025; Anthropic, 2025; Google, 2025). However, despite these advancements, even the most powerful LLMs still cannot reliably self-verify their own outputs (Lu et al., 2025; Stechly et al., 2025).

### 5.2. LLMs as Verifier

Verifiers play a crucial role in guiding LLMs toward better generations and enabling effective test-time scaling (Zhong et al., 2025; Yu et al., 2025b; Snell et al., 2024). Existing verifiers are typically either (1) discriminative (Liu et al., 2025a; 2024), producing scalar scores to rank candidate responses, or (2) generative (Zhang et al., 2025b; Mahan et al., 2024; Liu et al., 2025c), producing textual judgments or reward signals. With the success of RLVR in training stronger generators, increasing attention has been paid to generative verifiers due to their better generalization ability. LLM verifiers typically produce natural language rationales or textual judgments, which improve transparency and evaluation reliability. Correspondingly, training methods for LLM verifiers have evolved from supervised fine-tuning (SFT) to direct preference optimization (DPO) (Chen et al., 2025a; Zhang et al., 2025b; Liu et al., 2025c), and more recently to RLVR (Chen et al., 2025b; Yu et al., 2025d), inspired by advances in reasoning-oriented models. Despite these advances, how training as verifiers influences the model itself as a generator remains largely underexplored.

### 5.3. Joint Training of Generator and Verifier

Recently, several works have begun to explore incorporating verification signals into generator training. Among them, (Chen et al., 2025c) shows that collecting correctness signals on external models' outputs and training LLMs via imitation learning with fixed templates can shorten generated responses, albeit sometimes at the cost of slightly degraded generation performance. Other works (Liu et al., 2025b; Zhang et al., 2025a; Wang et al., 2025b) propose to jointly train generation and verification within the same training step, where verification is scaled as an auxiliary signal while the overall training dynamics remain dominated by the generation objective.

Different from these approaches, we notice that rewarding self-verification alone is sufficient to obtain a generator with performance comparable to standard generation training, while producing better reasoning traces. We further explore integrating self-verification into generation training by formulating a multi-task reinforcement learning framework, where generation and self-verification are optimized as two

decoupled but complementary objectives. We provide a more detailed discussion of our differences from prior work in Appendix B.

## 6. Conclusion

In this work, we investigate the asymmetry between generation and self-verification in large language models and show that improving generation does not naturally lead to better self-verification, even on the same task. More interestingly, we identify the reverse direction of this asymmetry: learning to self-verify alone can significantly improve generation performance. This finding challenges the common view of verification as merely an auxiliary component and highlights its role as a powerful training signal. Building on this insight, we further explore integrating self-verification into generation training by formulating a multi-objective reinforcement framework. Extensive experiments demonstrate that explicit self-verification training consistently improves problem-solving performance, produces more efficient and effective reasoning traces, and enables effective test-time scaling. Looking ahead, we believe that verification has untapped potential to improve generation, through designed verification tasks, more principled integration of verification and generation objectives, and more efficient training strategies. Exploring these directions is beyond the scope of the current work, and we leave them for future research.

## Impact Statement

Our findings suggest that strengthening self-verification in large language models can fundamentally change how these systems reason and generate responses. By showing that learning to self-verify not only improves reliability but also enhances generation efficiency, this work contributes to a better understanding of the interaction between reasoning, verification, and generation in modern language models. These insights have broader implications for the development and deployment of AI systems, especially in scenarios where correctness, robustness, and controllability are critical. Improving a model's ability to assess its own outputs may help reduce spurious reasoning steps, increase transparency, and mitigate certain classes of errors in real-world applications. At the same time, more powerful self-verification capabilities also raise new questions about how such systems should be evaluated, monitored, and governed, particularly when they are used in high-stakes or decision-critical settings. Understanding and carefully managing these dynamics is therefore important for the responsible use of large language models in practice.

## Limitation

Although this work provides an encouraging analysis of the asymmetry between generation and self-verification and demonstrates the effectiveness of learning to self-verify, it still has several limitations. First, introducing an additional self-verification objective into generation training inevitably incurs extra computation, including additional inference and optimization costs. Second, although our experiments cover models of different parameter sizes, they are still limited in scale. Due to computational constraints, we do not explore whether the same phenomena and benefits continue to hold for larger models. Also, despite its effectiveness, we only explore a limited set of ways to combine generation and self-verification, as well as a single form of verification task. More diverse verification formulations and tighter coupling paradigms between generation and verification may further push the performance ceiling. Moreover, our study focuses primarily on mathematical reasoning benchmarks. While the proposed framework is conceptually general, it remains an open question whether the same asymmetry and the benefits of learning to self-verify will hold in other domains, such as planning and multimodal reasoning. Finally, the current multi-task training schedule (e.g., stage-wise or alternating) is manually designed and heuristic. A more principled or adaptive strategy for balancing generation and self-verification objectives remains an interesting direction for future work. We further clarify the intended RL-compatible scope of our framework and discuss limitations of outcome-level verification rewards in Appendix A.6.

## Acknowledgment

This research is supported by the New Generation Artificial Intelligence-National Science and Technology Major Project (2025ZD0123302), the National Research Foundation, Singapore, under its National Large Language Models Funding Initiative (AISG Award No: AISG-NMLP-2024-002). Any opinions, findings and conclusions or recommendations expressed in this material are those of the authors and do not reflect the views of National Research Foundation, Singapore.

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

*Table 5.* General reasoning results with Qwen2.5-Instruct models. We evaluate on GPQA, MMLU-Redux, ZebraLogic, and IFEval, and report average accuracy and average token length.

| Method | Scale | General Reasoning Benchmarks | | | | Average | |
|--------|-------|------|------------|------------|--------|------|--------|
| | | GPQA | MMLU-Redux | ZebraLogic | IFEval | Acc↑ | Tokens↓ |
| **Qwen2.5-1.5B-Instruct** | | | | | | | |
| Generate | 1.5B | 23.2 | 53.1 | 25.9 | 50.3 | 38.1 | 2309 |
| Self-Verify | 1.5B | **26.8** | **56.6** | **29.3** | **54.7** | **41.9** | **1413** |
| **Qwen2.5-3B-Instruct** | | | | | | | |
| Generate | 3B | 25.8 | 62.0 | 20.1 | 65.8 | 43.4 | 2883 |
| Self-Verify | 3B | **29.8** | **65.8** | **24.4** | **68.5** | **47.1** | **2169** |
| **Qwen2.5-7B-Instruct** | | | | | | | |
| Generate | 7B | 34.9 | 69.7 | 35.2 | 77.8 | 54.4 | 1630 |
| Self-Verify | 7B | **35.4** | **74.7** | **35.6** | **81.0** | **56.7** | **1470** |

*Table 6.* Code generation results with Qwen2.5-7B-Instruct. The model is trained on APPS and evaluated on HumanEval and MBPP.

| Method | HumanEval | MBPP | Avg↑ |
|--------|-----------|------|------|
| Generate | 85.4 | 75.9 | 80.6 |
| Self-Verify | **86.6** | **77.8** | **82.2** |

# A. Additional Experiments and Analyses

This appendix provides additional experiments and analyses that further support the main findings in the paper. We first evaluate whether the observed effect of self-verification generalizes beyond the main mathematical reasoning setting. We then analyze the role of the verification data construction pipeline, the importance of on-policy samples, the stability of the integrated training results, and the intended scope of our method.

## A.1. Generalization Beyond the Main Math Setting

In the main paper, we primarily evaluate self-verification training on mathematical reasoning benchmarks. To examine whether the observed trend is specific to this setting, we further conduct experiments on general reasoning, code generation, different model families, and a stronger math-specialized backbone. Overall, the results show a consistent pattern: self-verification training achieves competitive or better accuracy than generation training, while often producing shorter responses.

**General reasoning with Qwen2.5.** We evaluate Qwen2.5-Instruct models on four general reasoning benchmarks, including GPQA, MMLU-Redux, ZebraLogic, and IFEval. As shown in Table 5, self-verification training consistently improves average performance across model scales and reduces the average number of tokens.

**Code generation.** We further evaluate Qwen2.5-7B-Instruct on code generation. The model is trained on APPS and evaluated on HumanEval and MBPP. As shown in Table 6, self-verification training improves both HumanEval and MBPP performance.

**Different model family: Llama3.2.** To further examine whether the finding is tied to the Qwen model family, we train Llama3.2 models under the same comparison. Table 7 reports mathematical reasoning results, and Table 8 reports general reasoning results. The gains are smaller than those observed on Qwen2.5, but the overall trend remains consistent: self-verification training is competitive in accuracy and often reduces token usage.

**Stronger math-specialized backbone.** We also evaluate Qwen2.5-Math-1.5B-Instruct to test whether the observed effect still holds on a stronger math-specialized model. As shown in Table 9, self-verification training remains competitive with

*Table 7.* Mathematical reasoning results with Llama3.2 models. We report task accuracy on five benchmarks, average accuracy, and average token length.

| Method | Scale | Mathematical Reasoning Benchmarks | | | | | Average | |
|---|---|---|---|---|---|---|---|---|
| | | Math500 | Minerva | Olympiad | AIME24 | AIME25 | Acc↑ | Tokens↓ |
| **Llama3.2-3B** | | | | | | | | |
| Generate | 3B | **49.6** | 12.3 | **18.3** | 11.7 | **2.5** | 18.9 | 1169 |
| Self-Verify | 3B | 48.1 | **15.0** | 16.9 | **15.0** | 1.7 | **19.3** | **922** |
| **Llama3.2-1B** | | | | | | | | |
| Generate | 1B | 24.7 | **4.8** | 6.3 | **2.5** | **0.0** | 7.7 | 1584 |
| Self-Verify | 1B | **26.9** | 3.6 | **6.5** | 2.3 | **0.0** | **7.8** | **1482** |

*Table 8.* General reasoning results with Llama3.2 models. We evaluate on GPQA, MMLU-Redux, ZebraLogic, and IFEval.

| Method | Scale | General Reasoning Benchmarks | | | | Average | |
|---|---|---|---|---|---|---|---|
| | | GPQA | MMLU-Redux | ZebraLogic | IFEval | Acc↑ | Tokens↓ |
| **Llama3.2-3B** | | | | | | | |
| Generate | 3B | 25.4 | 58.5 | 6.7 | **69.9** | 40.1 | 1426 |
| Self-Verify | 3B | **25.8** | **59.0** | **6.9** | **69.9** | **40.4** | **850** |
| **Llama3.2-1B** | | | | | | | |
| Generate | 1B | **26.3** | **39.3** | **0.5** | 57.1 | **30.8** | 1436 |
| Self-Verify | 1B | 26.1 | 37.0 | 0.1 | **57.9** | 30.3 | **1397** |

generation training and substantially reduces token usage.

## A.2. Ablation on Verification Data Construction

Our self-verification training pipeline constructs verification examples from the model's own rollouts and applies three simple post-processing operations: diversity control, label balancing, and filtering. To understand the role of these components, we remove each component in turn and train Qwen2.5-1.5B-Instruct under the same setting.

The results in Table 10 show that label balancing and filtering are important for stable self-verification training. Removing label balancing introduces class imbalance in the binary verification task, which may bias optimization toward trivial judgments. Removing filtering introduces invalid or noisy trajectories, leading to lower accuracy and longer responses. In contrast, removing diversity control has a smaller effect, consistent with its role as a fairness-oriented sampling constraint rather than a source of stronger supervision. These results suggest that the observed gains are not due to privileged supervision, but require basic task-validity controls to make the self-verification objective well-posed.

## A.3. On-Policy Samples Are Important for Self-Verification

Our main framework trains the model to verify its own on-policy generations. To test whether on-policy sampling matters, we construct an off-policy verification dataset using the same set of queries, while replacing the model's own rollouts with responses collected from other models, including Qwen3-1.7B, Qwen3-8B, and s1.1-7B. We then train Qwen2.5-1.5B-Instruct on this off-policy verification data and compare it with the standard self-verification setting.

As shown in Table 11, off-policy verification performs substantially worse than the on-policy self-verification setting. This result indicates that self-verification is not equivalent to training a generic verifier on arbitrary model outputs. Instead, learning to verify the model's own generations appears important, because the verification task is aligned with the model's current reasoning distribution and failure patterns.

*Table 9.* Results on Qwen2.5-Math-1.5B-Instruct. We compare generation training and self-verification training on mathematical reasoning benchmarks.

| Method | Benchmarks | | | | | Average | |
|---|---|---|---|---|---|---|---|
| | **Math500** | **Minerva** | **Olympiad** | **AIME24** | **AIME25** | Acc↑ | Tokens↓ |
| Generate | 70.5 | **18.8** | 35.1 | **9.2** | **5.8** | 31.7 | 640 |
| Self-Verify | **71.1** | **18.8** | **35.3** | 8.3 | **5.8** | **31.9** | **483** |

*Table 10.* Ablation on the verification data construction pipeline with Qwen2.5-1.5B-Instruct. We report average accuracy and average token length.

| Setting | Avg Acc↑ | Avg Tokens↓ |
|---|---|---|
| Origin | **21.7** | 1227 |
| w/o Diversity | 21.3 | **1175** |
| w/o Balance | 20.3 | 2029 |
| w/o Filtering | 20.2 | 1958 |

### A.4. Stability Across Repeated Runs

To examine the stability of the integrated training results, we conduct three independent runs and report the mean and standard deviation. We compare standard generation training with Verify-Alter, our alternating training strategy.

The improvements of Verify-Alter consistently exceed the observed standard deviation, suggesting that the gains are stable rather than being caused by random training fluctuations.

### A.5. Why Decoupled Training Helps

In Section 4, we compare our decoupled training strategies with Mixed-Train, which optimizes generation and verification objectives within the same training step. Although verification is beneficial to generation, directly mixing the two objectives can be suboptimal. We hypothesize that this is caused by three factors.

First, *optimization interference* may arise because gradients from generation and verification can point in different directions. The generation objective encourages the model to solve the task, whereas the verification objective encourages the model to judge a given response. Directly optimizing both objectives in the same step may weaken one or both signals. Second, *task imbalance* may affect optimization because generation and verification have different intrinsic difficulty. Verification is a binary decision task, where random guessing already gives a non-trivial baseline, while generation requires producing a full correct solution. Aggregating their normalized advantages at the same scale may therefore bias the learning dynamics. Third, *shortcut optimization* may occur when the model improves the joint reward without improving reasoning, for example by producing incorrect solutions while correctly judging them as wrong.

This explains why weaker Mixed-Train performance does not contradict the claim that self-verification helps generation. Instead, it suggests that the way the two objectives are integrated matters. Our decoupled strategies first allow self-verification to be learned as a stable and independent capability, and then transfer this capability back to generation.

### A.6. Scope and Limitations

Our framework assumes that the underlying task can be optimized with reinforcement learning, i.e., that some form of reward signal is available for the base task. Self-verification in our framework is not intended to eliminate the need for task rewards. Instead, it serves as an auxiliary or alternating training objective under the same RL-compatible setting. Therefore, domains where the base task itself cannot provide any usable reward signal are outside the intended scope of the current method and may require a different formulation.

In addition, our current implementation uses outcome-level binary verification rewards. This reward is scalable, but it does not explicitly supervise the quality of the model's verification rationale. As a result, a model may receive a positive reward

*Table 11.* Comparison between on-policy and off-policy verification training with Qwen2.5-1.5B-Instruct. Self-Verify uses the model's own rollouts, while Off-policy Verify uses responses from other models.

| Method | Mathematical Reasoning Benchmarks | | | | | | Average | |
|---|---|---|---|---|---|---|---|---|
| | AMC23 | Minerva | Olympiad | Math500 | AIME24 | AIME25 | Acc↑ | Tokens↓ |
| Self-Verify | **33.0** | **14.0** | **22.2** | **54.6** | **4.8** | 1.3 | **21.7** | **1227** |
| Off-policy Verify | 28.7 | 11.1 | 18.7 | 51.6 | 3.3 | **1.5** | 19.2 | 1228 |

*Table 12.* Repeated-run results over three independent trials. We report average accuracy with standard deviation.

| Model | Generate | Verify-Alter | $\Delta$ |
|---|---|---|---|
| Qwen2.5-1.5B-Instruct | $20.3 \pm 0.4$ | $\mathbf{22.6} \pm 0.5$ | +2.3 |
| Qwen2.5-3B-Instruct | $28.4 \pm 0.3$ | $\mathbf{29.7} \pm 0.3$ | +1.3 |
| Qwen2.5-7B-Instruct | $38.8 \pm 0.5$ | $\mathbf{40.6} \pm 0.4$ | +1.8 |

when the final judgment is correct, even if its intermediate verification reasoning contains errors. In our manual analysis with a stronger external evaluator, we observe that a small portion of verification traces still contain reasoning or factual errors. This suggests that future work could further improve self-verification training by introducing more fine-grained process supervision, harder verification tasks, or more diverse verification objectives.

# B. Discussion

In this section, we further clarify the main contributions and significance of our work, and discuss how it differs from existing studies on critique learning, self-verification, and joint optimization of generation and verification.

## B.1. Contributions and Significance

The central contribution of this work is not simply to introduce another verification-related training recipe, but to identify and systematically study an asymmetric relationship between generation and self-verification under RL training. Specifically, we show that improving generation on a task does not reliably improve the model's ability to verify its own answers on the same task. This suggests that self-verification does not naturally emerge as a by-product of stronger generation, even when the model is trained on the same task distribution.

More importantly, we find that the reverse direction behaves differently: learning to self-verify alone can transfer to better generation. This is a meaningful observation because self-verification is formulated as a binary decision task, whereas generation is an open-ended problem-solving task with a much larger output space. The fact that optimizing the model on such a reward-accessible verification task can improve open-ended generation indicates that self-verification can serve as an effective and transferable RL objective, rather than merely an auxiliary behavior that appears during reasoning.

The significance of this finding is further supported by the qualitative change in reasoning behavior. Models trained with self-verification not only achieve competitive or better final accuracy, but also produce substantially shorter reasoning traces. This suggests that explicit self-verification training can reduce redundant or ineffective checking behaviors and lead to more efficient reasoning. The corrupted-prefix experiments further show that self-verification-trained models are better at detecting and recovering from errors in intermediate reasoning. These results suggest that the benefit of self-verification is not limited to final answer accuracy, but also affects how the model reasons.

Based on this observation, we further explore self-verification as a flexible and decoupled training objective. Instead of always coupling verification with generation in the same optimization step, self-verification can be used as a warm-up objective or interleaved with generation training. This opens a broader design space for RL post-training, where simpler but transferable auxiliary tasks can be optimized independently and then used to improve more complex generation behavior.

## B.2. Difference from Existing Work

Existing work has explored several forms of critique learning, self-verification, and joint training. Our work differs from these studies in both problem setting and methodology.

**Difference from critique fine-tuning.**    Several prior studies train models to critique or verify responses by using teacher-generated critique data (Wang et al., 2025c;e). A common pipeline is to collect responses from external models, use a stronger teacher model to generate critiques, and then train the policy model to imitate these critiques through supervised fine-tuning. This setting differs from ours in two important aspects.

First, the verification target is different. Critique fine-tuning typically trains the model to critique answers produced by other models, which is closer to training a generic verifier over off-policy responses. In contrast, our work studies self-verification over the model's own on-policy generations. This distinction is nontrivial: verifying one's own generations requires the model to assess and refine reasoning trajectories that follow its own current policy distribution and failure patterns. Our off-policy comparison further shows that replacing on-policy rollouts with responses from other models leads to worse performance, suggesting that on-policy self-verification is not equivalent to generic critique learning.

Second, the supervision source is different. Teacher-dependent critique learning relies on an external model to construct critique chains (Wang et al., 2025c;e), which increases data construction cost and may limit the training signal by the teacher's capability. In contrast, our framework does not require teacher-generated critique rationales. The correctness label is obtained from a verifier, and the model learns self-verification through RL rather than imitation. Therefore, our setting studies whether self-verification can be acquired as an intrinsic RL objective, rather than whether a model can imitate another model's critiques.

**Difference from joint generation-verification training.**    Another line of work integrates verification into generation training by treating verification as an auxiliary signal jointly optimized with generation (Liu et al., 2025b; Zhang et al., 2025c; Wang et al., 2025d). Our work differs in focus. Rather than assuming verification should be coupled with generation in the same training step, we explicitly ask whether self-verification itself can be treated as an independent and transferable objective.

This distinction matters because directly mixing generation and verification objectives can introduce optimization challenges. Generation and verification have different output spaces, difficulty levels, and reward structures. Generation requires producing a full solution, whereas verification is a binary decision task. Optimizing both objectives in the same step may therefore cause optimization interference, task imbalance, or shortcut behaviors. Our results with mixed training are consistent with this concern: direct objective mixing is not always the most effective way to exploit verification. In contrast, decoupled strategies allow the model to first acquire self-verification as a stable capability and then transfer this capability back to generation.

**Difference from efficient self-rewarding methods.**    Some prior work focuses on how to compute or use self-verification rewards more efficiently during training or inference (Yang et al., 2025b). This direction is largely orthogonal to ours. Our focus is not primarily on reducing the cost of reward computation, but on understanding what self-verification training contributes to reasoning and how it should be incorporated into RL training. In this sense, efficient self-rewarding mechanisms could potentially be combined with our framework, but they do not address the same research question.

**Why the RL setting is important.**    The distinction between SFT-based critique learning and RL-based self-verification is also important. Supervised fine-tuning mainly encourages imitation of provided critique traces, and may primarily teach surface-level critique patterns. In contrast, RL optimizes the model through outcome-level feedback on its own sampled behavior, making it more closely tied to the model's current policy and error distribution. This difference is central to our setting: we study whether optimizing a model to verify its own on-policy generations can induce improvements that transfer to open-ended generation. This view is consistent with recent observations that SFT and RL can exhibit different generalization behaviors in post-training (Chu et al., 2025). Our results suggest that this RL-based formulation enables a form of capability transfer that is not captured by prior teacher-imitation or off-policy critique-learning settings.

Overall, our work contributes a distinct perspective: self-verification is not merely a helper module for generation, nor only a critique behavior to imitate from stronger models. Instead, it can be formulated as a decoupled RL objective with its own learning dynamics and transferable benefits for reasoning.

