# OpenReview forum: "Learning to Self-Verify Makes Language Models Better Reasoners"
_ICML.cc/2026/Conference — ICML 2026 regular_

### Official Review · Reviewer_3EWA · 2026-02-23

**Soundness:** 3
**Presentation:** 3
**Significance:** 2
**Originality:** 1
**Overall Recommendation:** 3
**Confidence:** 4

**Summary:**

This paper reveals a fact that training large language model (LLMs) to perform self-verification can not only improve its judgment ability, but also help to improve its problem-solving capability. Through experiments with Qwen2.5 series models on math reasoning, the authors validate this claim. Finally, the authors explore the effectiveness of jointly optimizing the generation and self-verification capabilities of LLMs during RL.

**Compliance With Llm Reviewing Policy:**

Affirmed.

**Final Justification:**

I have reviewed the authors’ latest response and appreciate their clarifications, which address some of my concerns. However, from my  personal perspective, the contributions and insights of this paper are not sufficiently novel and surprising (I have also read Reviewer RK28’s response, with which I only partially agree). After careful consideration, I have decided to raise my score to 3.

**Key Questions For Authors:**

Please refer to the **Weaknesses** part.

**Limitations:**

The authors have included the Limitations and Impact Statement sections.

**Strengths And Weaknesses:**

## Strengths

(1) The paper is well-written. The presentation is clear.

(2) The motivation is straightforward, and is backed with empirical results.

(3) The experimental results validate the claims well.

## Weaknesses

(1) In my opinion, **the contribution is limited and the paper does not provide new insights to the community.** Previous studies [1,2] have already revealed that training the self-verification/self-critique capabilities of LLMs only can also bring improvement to model' problem-solving abilities. Thus, **the main finding in this paper is not new**. Furthermore, jointly optimizing the generation and self-verification capabilities has also been studied in previous studies [3,4]. However, they are not compared in this work. Overall, I think the contribution is limited.

(2) The experiments are insufficient. All experiments are conducted on "Qwen2.5+Math" combo, which cannot convince me. I suggest the authors to include experiments on other backbones and domains to show the generalizability in the future version. The self-verification results in Table 2 and test-time scaling results in Table 3 are not promising compared with recent studies [3,4].


 ----

[1] Wang, Yubo, Xiang Yue, and Wenhu Chen. "Critique fine-tuning: Learning to critique is more effective than learning to imitate." COLM 2025

[2] Wang, Yubo, et al. "Unleashing the Reasoning Potential of LLMs by Critique Fine-Tuning on One Problem." EMNLP 2025

[3] Liu, Xiaoyuan, et al. "Trust, But Verify: A Self-Verification Approach to Reinforcement Learning with Verifiable Rewards." arxiv 2025

[4] Yang, Wenkai, et al. "Laser: Reinforcement learning with last-token self-rewarding." arxiv 2025

---

> ### Author Rebuttal · Authors · 2026-03-31
>
> > W1. No new finding and lack of baseline.
>
> We thank the reviewer for this opportunity to better clarify our contribution.
>
> Our key findings are:
>
> > **Generation and self-verification exhibit a clear asymmetry during RL training.**
> >  Improving generation on a task does not reliably improve self-verification, while improving self-verification alone can consistently transfer to better generation.
>
> We believe this is a meaningful finding because self-verification is a simpler binary decision task, yet optimizing only this task can still improve open-ended generation, which is a possibility that has been largely underexplored in prior works. Moreover, the gain is not limited to final accuracy: the trained model also exhibits **more effective and more efficient reasoning**, evidenced by lower token usage and stronger continuation performance from incorrect reasoning prefixes. Building on this finding, we further explore to treat self-verification as a flexible and decoupled training objective, e.g., as a warm-up stage or via alternating training, rather than coupling it with generation optimization as auxilary signal.
>
> **Difference from prior work**
>
> - [1,2]  train models to **imitate teacher-generated critiques** over external answers. They rely on teacher supervision and critique-chain construction, and are therefore different from our setting of **self-verification as an intrinsic training task**. In addition, as discussed in our response to Reviewer_RK28, verifying one’s own on-policy generations is fundamentally different from critiquing answers produced by another model.
> - [3] is the closest RL-based work, and we actually **do include it as a baseline** in our experiments in section 4. However, [3] treats verification as an **auxiliary signal jointly optimized with generation in the same training step**, whereas our work studies whether self-verification can be treated as a **decoupled objective** that itself serves as an effective and transferable training target.
> - [4] mainly studies a more efficient way to compute self-verification reward. This is largely **orthogonal** to our focus, which is on **what self-verification training contributes** and **how it could be incorporated into RL training**.
>
> **On Mixed-Train vs. Decoupled Training**
>
> We hypothesize that directly combining these two objectives within a single training step may introduce several issues:
>
> - **Optimization interference**: gradients from the verification objective may conflict with those from generation, weakening the effectiveness of both signals.
> - **Task imbalance**: the two tasks have inherently different difficulty (e.g., verification is a binary decision task where random guessing already achieves 50% accuracy). Aggregating their normalized advantages at the same scale may therefore introduce bias in optimization.
> - **Shortcut optimization**, where the model improves the joint reward without improving reasoning (e.g., generating incorrect solutions while receiving high verification reward by correctly judging all solutions as wrong).
>
> The relatively weaker performance of Mixed-Train in Table 4 further support these hypotheses .
>
>
> > W2. Experiments are insufficient
>
> To better illustrate the generality of our findings, we add experiments across different model families, parameter scales, and task domains.
>
> ## Qwen-2.5 + General Reasoning
> Beyond math, we evaluate Qwen2.5 on general reasoning benchmarks.
>
> |Method|Scale|GPQA|MMLU-Redux|ZebraLogic|IFEval|Avg|Token|
> |-|-|-|-|-|-|-|-|
> |Generate|1.5B|23.2|53.1|25.9|50.3|38.1|2309|
> |Self-Verify|1.5B|26.8|56.6|29.3|54.7|**41.9**|1413|
> |Generate|3B|25.8|62.0|20.1|65.8|43.4|2883|
> |Self-Verify|3B|29.8|65.8|24.4|68.5|**47.1**|2169|
> |Generate|7B|34.9|69.7|35.2|77.8|54.4|1630|
> |Self-Verify|7B|35.4|74.7|35.6|81.0|**56.7**|1470|
>
> ## Qwen-2.5 + Code
>
> We also add experiments on code. Specifically, we train Qwen2.5-7B-Instruct on APPS and evaluate on HumanEval and MBPP, two widely used code generation benchmarks:
>
> |Method|HumanEval|MBPP|Avg|
> |-|-|-|-|
> |Generate|85.4|75.9|80.6|
> |Self-Verify|86.6|77.8|**82.2**|
>
> ## Llama-3.2 + Math & General Reasoning
> Beyond Qwen, we also train Llama-3.2 models on same settings.
>
> |Method|Scale|Math500|Minerva|Olympiad|AIME24|AIME25|Avg|Token|
> |-|-|-|-|-|-|-|-|-|
> |Generate|3B|49.6|12.3|18.3|11.7|2.5|18.9|1169|
> |Self-Verify|3B|48.1|15.0|16.9|15.0|1.7|**19.3**|922|
> |Generate|1B|24.7|4.8|6.3|2.5|0.0|7.7|1584|
> |Self-Verify|1B|26.9|3.6|6.5|2.3|0.0|**7.8**|1482|
>
> |Method|Scale|GPQA|MMLU-Redux|ZebraLogic|IFEval|Avg|Token|
> |-|-|-|-|-|-|-|-|
> |Generate|3B|25.4|58.5|6.7|69.9|40.1|1426|
> |Self-Verify|3B|25.8|59.0|6.9|69.9|**40.4**|850|
> |Generate|1B|26.3|39.3|0.5|57.1|**30.8**|1436|
> |Self-Verify|1B|26.1|37.0|0.1|57.9|30.3|1397|
>
>
> The overall pattern is consistent: self-verification training is competitive or better in accuracy, and frequently achieves with fewer tokens. We thank the reviewer for prompting this broader evaluation, which we believe substantially strengthens the paper.

---

> > ### Author Rebuttal · Reviewer_3EWA · 2026-04-03
> >
> > In my view, this work primarily shifts the training setting from using verification/critique data generated by other models to using self-verification thoughts generated by the model itself. Methodologically, this does not constitute a fundamental difference, and the results are not surprising and novel to me.

---

> > > ### Author Response · Authors · 2026-04-05
> > >
> > > Thanks for continued engagement in the rebuttal discussion.
> > >
> > > We would like to clarify that our work is not merely "shifts the training setting from using verification data generated by other models to using self-verification thoughts generated by the model itself".
> > > Rather, our key finding is **an asymmetry between generation and self-verification during RL training**, which we believe is both underexplored and important.
> > >
> > > **One side of this asymmetry is that improving generation on a task does not improve self-verification.**
> > > Prior works have shown that LLMs can't reliably verify themselves and stronger generator do not make it stronger self-verifier. Our work takes one step further: we show that even when trained on the same batch of data, the model’s ability to verify its own answers on that same data does not improve. This suggests that **self-verification exhibits distinct and underexplored learning dynamics during RL training**, rather than emerging naturally from improved generation performance.
> > >
> > > **The other side of this asymmetry is that eliciting the model’s self-verification ability alone can transfer to better generation.**
> > > We believe the reviewer’s concern about our contribution may mainly stem from the fact that this part of our findings appears conceptually related to two prior works, since we both involve training verification-related capabilities. However, **our work differs fundamentally in both scope and methodology.**
> > >
> > > In particular, the prior works mentioned follow a shared training pipeline:
> > > - collect responses from other models,
> > > - use a teacher model to critique responses, and
> > > - train the policy model to imitate the teacher-generated critiques with SFT.
> > >
> > > As Reviewer RK28 points out, these methods are SFT-based. More importantly, what they optimize is **for the policy model to imitate how a stronger teacher critiques another model’s outputs**.
> > > By contrast, our work is fundamentally different from them.
> > >
> > > First, regarding the verification target, **the distinction between off-policy data (i.e., verify others) and on-policy self-verification is nontrivial**. As discussed in Reviewer RK28 Q1, off-policy critique data is much closer to training a verifier over external responses, rather than improving the model’s ability to assess and refine its own reasoning trajectories. Empirically, we find that such off-policy data performs substantially worse and even collapses early during training. In addition, collecting responses from different external models before training introduces additional reasoning cost and variability in data quality.
> > >
> > > Second, regarding the supervision pipeline, **our method does not rely on any external model**. We believe teacher-dependent training pipelines increase the cost of data construction, reduce scalability, and may inherently be limited by the upper bound of the teacher model, since the resulting training signal depends heavily on the quality of teacher-generated critique trajectories.
> > >
> > > More fundamentally, **eliciting a model to self-verify under RL is conceptually and methodologically different from training it to imitate a teacher model’s critiques**.
> > > RLVR has increasingly enabled new forms of capability acquisition that differ fundamentally from imitation-based learning.
> > > **As Reviewer RK28 also noted in the rebuttal acknowledgement**, these approaches differ in training pipelines and patterns. **This point is also consistent with Reviewer QaSq’s review and our discussion there**: SFT and RL exhibit fundamentally different dynamics.
> > >
> > > Beyond distinguishing from prior work, we would like to emphasize why this finding is significant from its methodological implications for RLVR.
> > > Unlike open-ended reasoning tasks such as math reasoning, which have a large answer space, self-verification is essentially a binary decision task. **We find it interesting that RL training on such a reward-accessible task alone can transfer to improved open-ended reasoning performance, which is under-explored in prior work.**
> > > Moreover, this improvement is not limited to accuracy: the model also exhibits more effective and efficient reasoning, evidenced by lower token usage and stronger continuation performance from incorrect prefixes.
> > >
> > > Regarding generality, we follow the reviewer’s suggestion, extending our evaluation to different model families and scales, domains, and OOD evaluations.
> > > Since the observed findings remains consistent, we believe this substantially strengthens our claim.
> > >
> > > Finally, based on this finding, we propose to treat self-verification as a flexible and decoupled training objective.
> > > **We believe this design space also has not been explored in prior work.** For the joint-training method mentioned, we have already included it as a baseline in our original paper and discussed them in detail during the rebuttal.
> > >
> > > We hope the above clarifications help better convey the originality and significance of our work, we would appreciate the reviewer reconsidering the assessment.

---

### Official Review · Reviewer_QaSq · 2026-03-09

**Soundness:** 3
**Presentation:** 3
**Significance:** 3
**Originality:** 3
**Overall Recommendation:** 4
**Confidence:** 4

**Summary:**

This paper examines a general domain of improving the reasoning capabilities of LLM by addressing the observed asymmetry between generation and self-verification. The authors demonstrate learning to self-verify significantly improves generation performance and reasoning efficiency. Extensive experiments on mathematical reasoning benchmarks show that their proposed method outperforms standard generation-only training in both accuracy and token usage, while also enabling effective test-time scaling.

**Compliance With Llm Reviewing Policy:**

Affirmed.

**Final Justification:**

The author has added more experiments other than mathematics, which has fully dispelled my doubts about the validity of their viewpoint in other fields. I will continue to maintain the evaluation of week accept.

**Key Questions For Authors:**

1. The specific experimental settings are not clearly described. For instance, regarding the training curves in Figure 1, I did not find the corresponding data in the subsequent text. Moreover, most experiments in this paper do not provide sufficient hyperparameters (e.g., the parameters related to GRPO), which makes it difficult to reproduce the experimental results.
2. RL experiments are often accompanied by significant uncertainty. Does replacing generation with “Alter verify” training yield a significant and stable performance improvement? Conducting multiple repeated experiments is an effective means to prevent random errors from compromising the reliability of the results.
3. I am very curious about the advantages of staged or alternating training compared to the mixed-train‘s co-training approach. In SFT tasks, it is rare to find works that demonstrate significant performance improvements by training different subtasks in separate stages or alternating training.

**Limitations:**

yes.

**Strengths And Weaknesses:**

Strengths:
1. This paper find an interesting asymmetry between generation and self-verification: While improving generation ability doesn’t necessarily enhance self-verification, learning to self-verify can significantly boost generation performance. and self-verification training leads to shorter reasoning traces, indicating more efficient problem-solving strategies.

2. The paper proposes a multi-task reinforcement learning framework that integrates self-verification into generation training, achieving consistent performance improvements.

Weaknesses:
1. The paper claims in its title that “Self-Verify Makes Language Models Better Reasoners,” implying a general improvement in reasoning capabilities. However, all empirical validations are conducted exclusively on mathematical reasoning benchmarks (e.g., AIME, MATH). While mathematical problems effectively demonstrate logical reasoning, the lack of experimental support from other domains (such as code generation and logical puzzles) renders the term “Reasoners” in the title overly broad. The authors should either refine the title to “Better Mathematical Reasoners” or include experiments on broader domains to demonstrate the generalizability of the proposed method.

2. A central thesis of this paper is that enhancing self-verification capabilities can, in turn, improve generation performance. The results in Table 4 demonstrate that the “Verify-Init” strategy (training on self-verification for the first 400 steps) indeed outperforms standard generation training. However, I am concerned by the observation that the `Mixed-Train` strategy, which directly combines verification and generation rewards, underperforms compared to the generation-only baseline on the Qwen2.5-7B-Instruct model. This seems contradictory to the core hypothesis that verification aids generation. Could the authors explain why a direct combination of objectives leads to performance degradation? Does this imply that there are inherent optimization conflicts between generation and verification objectives that can only be mitigated by the proposed decoupled training strategies?

3. While the authors mention computational constraints, I argue that selecting a more appropriate base model remains feasible within the available resources. The experiments are conducted using the general-purpose `Qwen2.5-Instruct` series rather than the stronger, domain-specific `Qwen2.5-Math`series. Stronger base models typically imply that different emergent behaviors or training dynamics may be triggered during the reinforcement learning process. Given that `Qwen2.5-Math` possesses higher baseline performance in mathematical reasoning, I am curious whether the proposed method remains robust under such “stronger base” conditions, or if its effectiveness relies on the model having relatively weaker initial reasoning capabilities?

---

> ### Author Rebuttal · Authors · 2026-03-31
>
> > W1. Whether add benchmarks or refine title.
>
> To better illustrate the generality of our findings to support the "reasoner" in title, we add experiments on new benchmarks.
>
> ## General Reasoning
> Beyond math, we evaluate Qwen2.5 on general reasoning benchmarks.
>
> |Method|Scale|GPQA|MMLU-Redux|ZebraLogic|IFEval|Avg|Token|
> |-|-|-|-|-|-|-|-|
> |Generate|1.5B|23.2|53.1|25.9|50.3|38.1|2309|
> |Self-Verify|1.5B|26.8|56.6|29.3|54.7|**41.9**|1413|
> |Generate|3B|25.8|62.0|20.1|65.8|43.4|2883|
> |Self-Verify|3B|29.8|65.8|24.4|68.5|**47.1**|2169|
> |Generate|7B|34.9|69.7|35.2|77.8|54.4|1630|
> |Self-Verify|7B|35.4|74.7|35.6|81.0|**56.7**|1470|
>
> ## Code
>
> We also add experiments on code. Specifically, we train Qwen2.5-7B-Instruct on APPS and evaluate on HumanEval and MBPP, two widely used code generation benchmarks:
>
> |Method|HumanEval|MBPP|Avg|
> |-|-|-|-|
> |Generate|85.4|75.9|80.6|
> |Self-Verify|86.6|77.8|**82.2**|
>
> **For results of Llama, please refer to our reponse to reviewer_RK28**
>
> The overall pattern is consistent: self-verification training is competitive or better in accuracy, and frequently achieves with fewer tokens. We thank the reviewer for prompting this broader evaluation, which we believe substantially strengthens the paper.
>
> > W2. Why a direct combination leads to performance degradation
>
> Mixed-Train naively optimizes two distinct objectives in the same step, which may introduce several issues:
> - **optimization interference**, where directly adding the two objectives produce conflicting gradients;
> - **task imbalance**, aggregating their normalized advantages at the same scale may introduce bias in optimization since generation and verification have different difficulty;
> - **shortcut optimization**, where the model improves the joint reward without improving reasoning (e.g., generating incorrect solutions while receiving high verification reward by correctly judging all solutions as wrong).
>
> This is exactly why we propose decoupled training strategies, allowing self-verification to be learned more stably and then transferred back to generation. In this sense, the weaker performance of Mixed-Train does not weaken our hypothesis; rather, it highlights that **how to jointly optimize complementary tasks is itself important**.
>
> > W3. No Results on Qwen2.5-Math
>
> To address this concern, we now additionally run experiments on Qwen2.5-Math-1.5B-Instruct:
>
> |Method|Math500|Minerva|Olympiad|AIME24|AIME25|Avg|Token|
> |-|-|-|-|-|-|-|-|
> |Generate|70.5|18.8|35.1|9.2|5.8|31.7|640|
> |Self-Verify|71.1|18.8|35.3|8.3|5.8|**31.9**|483|
>
> The overall trend remains consistent with the Instruct models.
>
> > Q1. Lack of details description of some experiments.
>
> For Figure 1, the curves correspond to the training runs that produce Table 1.
> For the GRPO setup, the main settings for Table 1 are already provided in Section 3.2, and Figure 1 uses the same configuration. Table 4 largely follows the same GRPO setup, except for additional settings (e.g., the Verify-Init schedule), which are described in Section 4.2.
>
>
> > Q2. Conduct multiple repeated experiments
>
> We run three independent trials and report mean ± std:
>
> |Model|Gen|Verify-Alter|Δ|
> |-|-|-|-|
> |Qwen2.5-1.5B-Instruct|20.3 ± 0.4|**22.6 ± 0.5**|+2.3|
> |Qwen2.5-3B-Instruct|28.4 ± 0.3|**29.7 ± 0.3**|+1.3|
> |Qwen2.5-7B-Instruct|38.8 ± 0.5|**40.6 ± 0.4**|+1.8|
>
> Verify-Alter consistently outperforms generation, and the gains exceed the variance, indicating stable improvements rather than random fluctuations.
>
> > Q3. Curious about the advantage of staged training and its connection about SFT tasks.
>
> The advantages of staged training over mixed co-training have already been partially discussed in our response to Weakness 2. Here, we further discuss why, in standard SFT settings, training on different subtasks often does not lead to significant performance gains, whereas such benefits may become more evident in our RL-based setting. We believe the main reason lies in the fundamentally different properties of SFT and RL. Recent work suggests that SFT and RL exhibit fundamentally different optimization behaviors: SFT tends to favor memorization and can degrade out-of-distribution generalization, while RL tends to preserve more generalizable behaviors and forget less [1]. As a result, when different subtasks are learned through SFT, the model may mainly absorb task-specific surface patterns without substantially improving transferable capabilities. In contrast, RL is more closely tied to outcome-level optimization and on-policy refinement, making it more likely for training on one subtask to induce improvements that transfer to related capabilities. This is also consistent with a growing line of RL post-training work that adopts staged or cooperative optimization instead of fully mixed co-training, especially when the training signals are heterogeneous [2,3].
>
> [1] *SFT Memorizes, RL Generalizes*
>
> [2] *Beyond Two-Stage Training: Cooperative SFT and RL (BRIDGE)*
>
> [3] *ReVisual-R1: Staged RL for Multimodal Reasoning*

---

> > ### Author Rebuttal · Reviewer_QaSq · 2026-04-03
> >
> > Thank you for the author's detailed explanation. The author has added more experiments other than mathematics, which has fully dispelled my doubts about the validity of their viewpoint in other fields. I will continue to maintain the evaluation of week accept.

---

> > > ### Author Response · Authors · 2026-04-05
> > >
> > > We sincerely thank you for the positive feedback. We are glad that our response regarding the additional experiments and discussions fully addressed your concerns. Thank you for your support.

---

### Official Review · Reviewer_RK28 · 2026-03-15

**Soundness:** 3
**Presentation:** 4
**Significance:** 3
**Originality:** 3
**Overall Recommendation:** 5
**Confidence:** 3

**Summary:**

This paper investigates the relationship between the generation and verification capabilities of LLMs. Generation is the ability to directly generate a solution to a reasoning task, while self-verification is the ability to tell whether a solution is correct or not. Specifically, the authors construct a verification task by sampling reasoning traces in an on-policy fashion and compute their labels through a rule-based verifier. On six math benchmarks, the authors found a asymmetry between these two capabilities, where improving generation doesn’t improve self-verification performance, but improving self-verification does improve generation performance. It is observed that training on self-verification not only achieves a comparable accuracy against standard generation training, but also reduces the number of tokens to solve the task. In addition, the authors show that training on self-verification can somehow improve test-time scaling, and can be integrated with standard generation training to further improve the performance.

**Compliance With Llm Reviewing Policy:**

Affirmed.

**Final Justification:**

The authors did a good job in showing that learning to self-verify also helps general reasoning tasks during the discussion. As my initial score is already positive, I would keep my recommendation.

**Key Questions For Authors:**

1. How important is it to train on on-policy samples? The self-verification training is a little bit different from training a verifier in common sense, where the model is trained on off-policy traces. If on-policyness is the key to successful self-verification, that would be an important insight.
2. What does the column Avg@16 correspond to? Is it the average of six benchmarks?
3. In Table 3, please bold the result of major voting on Minerva as well.
4. In Table 4, it’s better to include training solely on self-verification from Table 1 as a baseline. I notice that self-verification sometimes even outperforms verify-init and verify-alter. For example, self-verification for Qwen2.5-7B-Instruct on Math500 is 74.7, but verify-init is 74.0 and verify-alter is 72.9. Can the authors explain that?

**Limitations:**

The authors have discussed several limitations of their method in the appendix. I would recommend the authors to move the limitation section to the main paper.

**Strengths And Weaknesses:**

Soundness:

- Strengths: The authors did thorough experiments to reveal the asymmetric generalization between generation and self-verification. The comparison is fair enough and the results are solid.
- Weaknesses: Only math benchmarks and Qwen2.5 models are studied in this paper. It’s not very clear to me that this observation is general or specific to math. From my understanding, learning self-verification may be very hard for coding tasks (e.g. verify whether a combinatorial solution is optimal).

Presentation:

- Strengths: This paper is well written and easy to follow. The authors positioned the work in the context of existing studies on verification abilities and joint training of generation and verification.

Significance:

- Strengths: It is significant to the community that revealing the asymmetric generalization between generation and verification. This points a new direction for future works on improving LLM reasoning.

Originality:

- Strengths: To my understanding, showing the asymmetry between generation and verification is original.
- Weaknesses: Joint training of verification and generation has been studied in previous works [1, 2, 3], as cited in this paper. Compared to them, the stage-wise and alternating training methods in this paper don’t provide much stronger results (Table 4) nor new insights.

[1] Liu et al. Trust, But Verify: A Self-Verification Approach to Reinforcement Learning with Verifiable Rewards. arXiv 2025.

[2] Zhang et al. Incentivizing LLMs to Self-Verify Their Answers. NeurIPS 2025.

[3] Wang et al. From Solving to Verifying: A Unified Objective for Robust Reasoning in LLMs. arXiv 2025.

---

> ### Author Rebuttal · Authors · 2026-03-31
>
> > W1. Only Qwen 2.5 and Math are studied.
>
> We thank the reviewer for this helpful suggestion. To better illustrate the generality of our findings, we add experiments across different model families, parameter scales, and task domains.
>
> ## Qwen-2.5 + General Reasoning
> Beyond math, we evaluate Qwen2.5 on general reasoning benchmarks.
>
> |Method|Scale|GPQA|MMLU-Redux|ZebraLogic|IFEval|Avg|Token|
> |-|-|-|-|-|-|-|-|
> |Generate|1.5B|23.2|53.1|25.9|50.3|38.1|2309|
> |Self-Verify|1.5B|26.8|56.6|29.3|54.7|**41.9**|1413|
> |Generate|3B|25.8|62.0|20.1|65.8|43.4|2883|
> |Self-Verify|3B|29.8|65.8|24.4|68.5|**47.1**|2169|
> |Generate|7B|34.9|69.7|35.2|77.8|54.4|1630|
> |Self-Verify|7B|35.4|74.7|35.6|81.0|**56.7**|1470|
>
> ## Qwen-2.5 + Code
>
> We also add experiments on code. Specifically, we train Qwen2.5-7B-Instruct on APPS and evaluate on HumanEval and MBPP, two widely used code generation benchmarks:
>
> |Method|HumanEval|MBPP|Avg|
> |-|-|-|-|
> |Generate|85.4|75.9|80.6|
> |Self-Verify|86.6|77.8|**82.2**|
>
> ## Llama-3.2 + Math & General Reasoning
> Beyond Qwen, we also train Llama-3.2 models on same settings.
>
> |Method|Scale|Math500|Minerva|Olympiad|AIME24|AIME25|Avg|Token|
> |-|-|-|-|-|-|-|-|-|
> |Generate|3B|49.6|12.3|18.3|11.7|2.5|18.9|1169|
> |Self-Verify|3B|48.1|15.0|16.9|15.0|1.7|**19.3**|922|
> |Generate|1B|24.7|4.8|6.3|2.5|0.0|7.7|1584|
> |Self-Verify|1B|26.9|3.6|6.5|2.3|0.0|**7.8**|1482|
>
> |Method|Scale|GPQA|MMLU-Redux|ZebraLogic|IFEval|Avg|Token|
> |-|-|-|-|-|-|-|-|
> |Generate|3B|25.4|58.5|6.7|69.9|40.1|1426|
> |Self-Verify|3B|25.8|59.0|6.9|69.9|**40.4**|850|
> |Generate|1B|26.3|39.3|0.5|57.1|**30.8**|1436|
> |Self-Verify|1B|26.1|37.0|0.1|57.9|30.3|1397|
>
> The overall pattern is consistent with: self-verification training is competitive or better in accuracy, and frequently achieves with fewer tokens. We thank the reviewer for prompting this broader evaluation, which we believe substantially strengthens the paper.
>
> > W2. Compared to joint training methods.
>
> For joint training, we hypotheses that direct joint optimization may introduce:
>
> - **optimization interference**, where directly adding the two objectives produce conflicting gradients;
> - **task imbalance**, aggregating their normalized advantages at the same scale may introduce bias in optimization since generation and verification have different difficulty;
> - **shortcut optimization**, where the model improves the joint reward without improving reasoning (e.g., generating incorrect solutions while receiving high verification reward by correctly judging all solutions as wrong).
>
> The relatively weaker performance of Mixed-Train in Table 4 is consistent with these hypotheses. Therefore, our contribution is not only a new training recipe, but also the finding that self-verification can be more effectively exploited as a decoupled yet transferable RL objective, which is underexplored in prior work.
>
> > Q1. How important is it to train on on-policy samples?
>
> We thank the reviewer for this insightful question. We agree that the role of on-policy samples is an important issue.
>
> To examine this, we construct an off-policy verification dataset using the same set of queries, but replacing the model’s own rollouts with responses collected from other models (Qwen3-1.7B, Qwen3-8B, s1.1-7B). We then compare it against our standard Self-Verify setting on Qwen2.5-1.5B-Instruct:
>
> |Method|AMC23|Minerva|Olympiad|Math500|AIME24|AIME25|Avg|Token|
> |-|-|-|-|-|-|-|-|-|
> |Self-Verify|33.0|14.0|22.2|54.6|4.8|1.3|**21.7**|1227|
> |Off-policy Verify|28.7|11.1|18.7|51.6|3.3|1.5|19.2|1228|
>
> We find that on-policy samples are indeed important: off-policy verification performs substantially worse than Self-Verify, and in our experiments collapses early in training. This result suggests that our self-verification framework is fundamentally different from prior works. We thank the reviewer for pointing out this important distinction.
>
> > Q2. Is Avg@16 the average of six benchmarks?
>
> Yes.
>
> > Q3. Lack of bold in Table 3.
>
> We will revise it accordingly.
>
> > Q4. Table presentation issue and explanation of sometimes higher performance of self-verification-only.
>
> We thank the reviewer for this suggestion. We agree that including the self-verification-only results from Table 1 in Table 4 would make the comparison more complete, and will add them in the revision.
>
> Regarding why verify-init or verify-alter does not always outperform verify-only, we believe this is consistent with our main finding: self-verification itself is already a strong training objective, and can sometimes outperform direct generation optimization (as also shown in Table 1).
>
> The goal of verify-init and verify-alter is not to consistently exceed verify-only, but to show that self-verification can be flexibly integrated into generation training (e.g., as a warm-up or interleaved objective) while still improving generation. These variants therefore highlight the transferability and compositionality of self-verification, rather than aiming to dominate verify-only in all settings.

---

> > ### Author Rebuttal · Reviewer_RK28 · 2026-04-04
> >
> > Thanks the authors for their proper response. It's great to see that learning to self-verify also helps general reasoning tasks. I suggest the authors including these new experiments in their final version. As my rating is already positive, I would like to keep the same rating.
> >
> > While Reviewer 3EWA argued that this paper has limited contributions, I checked the papers they mentioned and found they are based on SFT. From my understanding, generalization of SFT and RL could be very different, and what demonstrated in this paper remains significant to the community.

---

> > > ### Author Response · Authors · 2026-04-05
> > >
> > > We sincerely thank the reviewer for the valuable and professional feedback throughout the review and rebuttal process. We truly appreciate your thoughtful engagement with our work, and we believe your comments have helped improve the quality and clarity of the paper. We are also very encouraged that the core contribution of our work has been recognized. Thank you again for your support and constructive assessment.

---

### Official Review · Reviewer_mXQe · 2026-03-24

**Soundness:** 2
**Presentation:** 3
**Significance:** 3
**Originality:** 2
**Overall Recommendation:** 3
**Confidence:** 4

**Summary:**

This paper studies the asymmetry between generation and self-verification in LLMs. It shows empirically that improving generation does not improve verification ability, while training models to self-verify can improve generation performance. Building on this observation, the authors propose (1) a self-verification training pipeline using RL with verifiable rewards, and (2) a multi-task RL framework that jointly optimizes generation and verification via stage-wise initialization and alternating training. Experiments on mathematical reasoning benchmarks show that self-verification training achieves comparable or better accuracy with significantly shorter reasoning traces and improves test-time scaling.

**Compliance With Llm Reviewing Policy:**

Affirmed.

**Final Justification:**

The reviewers have partially addressed my concerns (e.g., W1, W3, W5). I'm still concerned with limited novelty, the main finding already exists in other literature. Using RLVR to train LLMs to self-verify is also not new. The main difference here is separating the training of generation and verification with engineered tricks while the results are not convincing enough. I've read other reviews and comments but still decide to keep weak reject.

**Key Questions For Authors:**

How would this approach extend to domains without verifiable rewards? If it does not generalize, the scope of the contribution is much narrower than implied.

Why does verification improve generation but not vice versa? Can you provide any mechanistic or theoretical explanation (e.g., credit assignment, representation learning, or implicit search)?

**Strengths And Weaknesses:**

Strength:

This paper explores what role verification plays in reasoning models, especially in the RLVR paradigm. The empirical observation of a directional asymmetry (verification to generation helps, but not vice versa) is interesting and, if robust, could have meaningful implications for training strategies. This paper evaluates across multiple model sizes and benchmarks, and includes analysis such as token efficiency and test-time scaling. The finding that self-verification reduces reasoning length substantially while maintaining accuracy is particularly noteworthy.

Weakness：

(1) The central claim that self-verification training improves generation is not convincingly isolated. It is unclear whether the gains come from verification as a capability or from better curated training data.

(2) Second, the method is highly ad hoc and engineering-heavy. The verification dataset construction includes multiple heuristics (discarding all-wrong samples, enforcing label balance, diversity sampling), none of which are theoretically justified. These choices could significantly influence results, yet no ablation is provided.

(3) Besides, the claimed “asymmetry” lacks deeper explanation. The paper documents the phenomenon but does not provide a theoretical or mechanistic account. Without this, it is unclear whether the result is fundamental or an artifact of the training setup.

(4) The binary verification reward is insufficient to supervise the quality of the model’s judgment. A model can produce incorrect verification reasoning while still receiving a positive reward if the final verdict matches the ground truth.

(5) All experiments are conducted on math reasoning benchmarks, where rule-based verifiers are available. It is unclear whether the findings generalize to domains without verifiable rewards (e.g., open-ended reasoning, coding without unit tests, or real-world tasks).

---

> ### Author Rebuttal · Authors · 2026-03-31
>
> > W1 & W2. The gain is driven by “better curated” training data and lack of ablation.
>
> We strictly compare generation training and self-verification training under a controlled setting. Training data of both tasks are constructed from the **same rollout pool**, and the key difference only lies in the **training objective**, rather than access to better supervision. Also, current self-verification data is not "carefully curated" as it is directly derived from the model’s own rollouts, with only minimal pre-processing needed to make verification training feasible. Specifically, since each query produces multiple rollouts, we downsample the rollout pool so that generation and self-verification are trained on the same set of queries with the same batch size. Besides, "verification as a capability" has been supported by our experiments in section 3.4.
>
> To make the role of each design choice clearer, we now add an ablation study on Qwen2.5-1.5B-Instruct, where we remove each component in turn:
>
> |Setting|Avg Acc|Avg Token|
> |-|-|-|
> |Origin |21.7|1227|
> |w/o Diversity|21.3|1175|
> |w/o Balance|20.3|2029|
> |w/o Filtering|20.2|1958|
>
> Removing label balancing (w/o Balance) introduces class imbalance, which biases optimization and may encourage reward hacking. Removing format filtering (w/o Filtering) introduces noisy trajectories. Both lead to lower accuracy and longer responses. In contrast, removing the one-sample-per-query restriction (w/o Diversity) has only a limited effect.
>
> These results suggest that how verification data is constructed does matter, and we thank the reviewer for pointing this out. We believe this analysis provides useful guidance for designing more effective verification training in future work.
>
> > W3 & Q2. “Asymmetry” lacks explanation.
>
> We believe this asymmetry is supported by both prior literature and our empirical findings and is not "an artifact of the training setup".
>
> On the one hand, it is already observed that improving generation does not reliably improve self-verification. Prior work shows that using the same model as both generator and verifier often brings limited gains [1]. This has been linked to the generator–verifier gap [2] and the broader difficulty of scaling self-verification as naturally as generation. We observe a consistent pattern in our RL training: improvements in generation performance do not translate into proportional gains in self-verification ability.
>
> On the other hand, the reverse direction—improving self-verification improves generation—is the main finding of our work. We observe that long-chain generation implicitly depends on “self-reflection”, but many such steps are often ineffective [3]. Our hypothesis is that explicit self-verification training improves the quality of these internal checks. This can by validated by our experiments: the model shows stronger recovery under corrupted-prefix settings and achieves better performance with fewer tokens, suggesting reduced reliance on redundant verification behaviors.
>
> [1] Large Language Models Cannot Self-Correct Reasoning Yet
>
>
> [2] Mind the Gap: Examining the Self-Improvement Capabilities of Large Language Models
>
>
> [3] Can aha moments be fake? identifying true and decorative thinking steps in chain-of-thought
>
> > W4. Outcome reward is insufficient.
>
>  This is a known limitation of outcome-based RL. Our claim, however, is not that such rewards are perfect, but that self-verification training remains effective for improving generation under this scalable supervision. Importantly, we find that even optimizing a simple binary verification task with imperfect rewards can improve open-ended generation, which is the core finding of our work.
>
> To further examine this issue, we use a stronger external LLM as an auxiliary evaluator and find that about 8% of verification traces still contain reasoning or factual errors. This confirms the need for process supervision in RLVR, but also shows that the observed gains are achieved without perfect supervision, leaving room for future improvement via more fine-grained objectives. We believe one promising direction is to design more challenging and diverse verification tasks, which may reduce the impact of this issue.
>
> > Q1. Extend to domains without verifiable rewards.
>
> Our framework is built on the assumption that the generation task itself is trainable with RL, i.e., that some form of reward signal is available for the underlying task. In this sense, self-verification in our framework is not intended to eliminate the need for the task reward, but rather to provide insight and serve as an alternating or auxiliary training objective under the same RL setting.
>
> Therefore, domains where the base task itself cannot be optimized with RL due to the absence of any reward signal fall outside the intended scope of our method. We believe evaluating such settings would require a fundamentally different problem formulation, rather than a straightforward extension of the current framework.

---

> > ### Author Rebuttal · Reviewer_mXQe · 2026-04-04
> >
> > Thank you for the rebuttal. The added ablation is helpful, but it does not change my overall assessment or score.
> >
> > My main concern remains that the paper’s gains are still hard to attribute cleanly to self-verification as a capability rather than to the particular data-construction and training recipe. In fact, the new ablation partly reinforces this concern. Removing balancing or filtering causes noticeable drops, which suggests that these heuristics are not peripheral details but important drivers of performance. At the same time, the overall gains are still fairly modest.
> >
> > I also still do not find the work sufficiently novel. The paper provides an interesting empirical observation, but the overall recipe remains fairly engineering-heavy. Finally, the scope remains narrow. All experiments are still in math reasoning with rule-based verifiers, so the broader implications remain unclear.

---

> > > ### Author Response · Authors · 2026-04-05
> > >
> > > Thank you for your continued engagement in the rebuttal discussion. We provide a point-by-point clarification below.
> > >
> > > > **"My main concern remains..."**
> > >
> > > To clarify this, we would like to highlight the **strict fairness of comparison**.
> > >
> > > Under the GRPO-based RLVR setup, each batch contains *B* questions, and each question is rolled out *N* times, yielding a rollout pool of size *B × N*. Importantly, **both generation and self-verification are constructed from exactly this same rollout pool**, meaning that they share the same questions *q* and gold labels *l* in each batch.
> > >
> > > For standard generation training, we follow the usual RLVR setup. For self-verification training, the input consisting of a question *q* and the model’s own responses *r*. A key issue is then how to construct self-verification training examples from this pool. Using **all** *B × N* rollouts would be the most straightforward choice, but it would also make the comparison inherently unfair: for every *B* questions, generation would perform **one** optimization step, whereas self-verification would optimize over ***N* times more training instances**. Therefore, to preserve fairness, we sample only *B* examples in total, specifically **one response *r* for each question *q***. "Ensuring sampling one *r* for each *q*" is precisely the role of the **“diversity”** component: it is not introduced to provide stronger supervision, but to ensure fairness in the comparison.
> > >
> > > >**"The paper...engineering-heavy."**
> > >
> > > We do not view our pipeline as “engineering-heavy”, but rather as a set of **minimal and well-motivated controls** required to make self-verification training reasonale.
> > >
> > > Once we sample one response *r* per question *q*, the most natural strategy would be to sample uniformly at random. However, because *r* is generated by the model itself, some responses are clearly invalid. Our **filtering** component simply removes responses with **format reward = 0**, i.e., those from which the final answer cannot even be extracted. Likewise, our **balancing** component follows naturally from the fact that self-verification is a **binary decision task**, where skewed label distributions can easily bias training or encourage trivial reward hacking. We therefore enforce a simple **1:1 positive/negative ratio**.
> > >
> > > These two components are straightforward steps to ensure that the sampled rollout-derived data forms a **valid verification task**.
> > >
> > > >**"In fact, the new ablation..."**
> > >
> > > We agree that these components are indeed **drivers of performance**, but we do **not** believe this implies that the gains come from an overly engineered recipe. Rather, the ablation supports a narrower conclusion: **once self-verification is formulated as a trainable RL objective, minimal task-validity controls such as filtering and balancing are necessary to make that objective work as intended**.
> > >
> > > This interpretation is also consistent with the ablation itself: **w/o Diversity** has only a limited effect, matching its fairness-oriented role, while **w/o Filtering** and **w/o Balancing** lead to drops because they directly affect whether the self-verification task is well-posed and learnable. We therefore believe the fairest interpretation is that the observed gains arise from the **self-verification objective under a controlled setup**, rather than from privileged supervision or an overly engineered training recipe.
> > >
> > >
> > > >**"I also still do not find the work sufficiently novel."**
> > >
> > > For a more detailed clarification of novelty, we have provided a fuller discussion in our response to **Reviewer 3EWA**, and we would kindly invite the reviewer to refer to that discussion as well.
> > >
> > > In short, our key novelty lies in identifying a **previously underexplored asymmetry between generation and self-verification under RL training**: improving generation does not improve self-verification, while **optimizing self-verification, a binary decision task, alone can transfer to better generation**.
> > >
> > > >**"Finally, the scope remains narrow..."**
> > >
> > > We thank the reviewer for raising this point. In fact, during the rebuttal phase, we have already followed this suggestion and extended our evaluation along multiple axes, including **different model families and scales** (Llama3.2-1B, Llama3.2-3B, Qwen2.5-Math-1.5B), **different domains** (math and code), and **OOD evaluations** on more general reasoning settings. We have reported these results in rebuttal responses. For detailed numbers, please refer to our responses to the other reviewers.
> > >
> > > Importantly, the key findings remain **consistent across all settings**, which we believe substantially strengthens our claim. While we agree that the current paper is still centered on RLVR-compatible domains, we believe these additional results make the broader implications of the observed asymmetry significantly clearer.
> > >
> > > We hope the above clarifications help better convey the significance of our work, we would appreciate the reviewer reconsidering the assessment.

---

### Decision · Program_Chairs · 2026-04-30

**Decision:**

Accept (regular)

**Comment:**

This paper receives scores of 4, 3, 5, 3. This paper studies the asymmetry between generation and self-verification in language models under RL training. In the paper, the authors argue that learning to self-verify can improve generation quality and efficiency while also serving as a useful complementary training objective. The reviewers generally agreed that the paper is clearly written, technically solid, and provide interesting empirical findings with meaningful implications for LLM reasoning. In particular, there are several strong aspects: (a) the careful empirical study of asymmetry, (b) the evidence that self-verification can yield competitive generation performance with shorter reasoning traces, and (c) the exploration of decoupled training strategies for integrating verification into generation.

The discussion mainly focuses on originality, the role of the data construction pipeline, and the scope of the empirical validation. The reviews and rebuttal are extensive, with the additional experiments and clarifications being added and provided. The rebuttal substantially strengthened the paper by adding broader evaluations across additional backbones and domains, repeated trials, and clarifications regarding on-policy data, ablations, and the distinction from prior work. While some concerns remain about the depth of the mechanistic explanation, and the breadth of the claims beyond RLVR-compatible settings, these issues do not outweigh the paper’s overall soundness, clarity, and value added to the community.

Therefore, the recommendation is to accept this paper. For the camera-ready version, the authors are encouraged to incorporate the additional rebuttal experiments and ablations into the main paper and clarify the intended scope/limitations of the proposed method. The camera-ready version also needs to address the reviewers’ comments on reproducibility and the interpretation of the joint-training results.